# A Geometric Perspective on Optimal Representations for Reinforcement Learning

**Marc G. Bellemare**[1], **Will Dabney**[2], **Robert Dadashi**[1], **Adrien Ali Taiga**[1,3],
**Pablo Samuel Castro**[1], **Nicolas Le Roux**[1], **Dale Schuurmans**[1,4], **Tor Lattimore**[2], **Clare Lyle**[5]

## Abstract

We propose a new perspective on representation learning in reinforcement learning based on geometric properties of the space of value functions. We leverage this perspective to provide formal evidence regarding the usefulness of value functions as auxiliary tasks. Our formulation considers adapting the representation to minimize the (linear) approximation of the value function of all stationary policies for a given environment. We show that this optimization reduces to making accurate predictions regarding a special class of value functions which we call *adversarial value functions* (AVFs). We demonstrate that using value functions as auxiliary tasks corresponds to an expected-error relaxation of our formulation, with AVFs a natural candidate, and identify a close relationship with proto-value functions (Mahadevan, 2005). We highlight characteristics of AVFs and their usefulness as auxiliary tasks in a series of experiments on the four-room domain.

## 1 Introduction

A good representation of state is key to practical success in reinforcement learning. While early applications used hand-engineered features (e.g. Samuel, 1959), these have proven onerous to generate and difficult to scale. As a result, methods in representation learning have flourished, ranging from basis adaptation (Menache et al., 2005; Keller et al., 2006), gradient-based learning (Yu and Bertsekas, 2009), proto-value functions (Mahadevan and Maggioni, 2007), feature generation schemes such as tile coding (Sutton, 1996) and the domain-independent features used in some Atari 2600 game-playing agents (Bellemare et al., 2013; Liang et al., 2016), and nonparametric methods (Ernst et al., 2005; Farahmand et al., 2016; Tosatto et al., 2017). Today, the method of choice is deep learning. Deep learning has made its mark by showing it can learn complex representations of relatively unprocessed inputs using gradient-based optimization (Tesauro, 1995; Mnih et al., 2015; Silver et al., 2016).

Most current deep reinforcement learning methods augment their main objective with additional losses called *auxiliary tasks*, typically with the aim of facilitating and regularizing the representation learning process. The UNREAL algorithm, for example, makes predictions about future pixel values (Jaderberg et al., 2017); recent work approximates a one-step transition model to achieve a similar effect (François-Lavet et al., 2018; Gelada et al., 2019). The good empirical performance of distributional reinforcement learning (Bellemare et al., 2017) has also been attributed to representation learning effects, with recent visualizations supporting this claim (Such et al., 2019). However, while there is now conclusive empirical evidence of the usefulness of auxiliary tasks, their design and justification remain on the whole ad-hoc. One of our main contributions is to provides a formal framework in which to reason about auxiliary tasks in reinforcement learning.

We begin by formulating an optimization problem whose solution is a form of optimal representation. Specifically, we seek a state representation from which we can best approximate the value function of any stationary policy for a given Markov Decision Process. Simultaneously, the largest approximation

error in that class serves as a measure of the quality of the representation. While our approach may appear naive – in real settings, most policies are uninteresting and hence may distract the representation learning process – we show that our representation learning problem can in fact be restricted to a special subset of value functions which we call *adversarial value functions* (AVFs). We then characterize these adversarial value functions and show they correspond to deterministic policies that either minimize or maximize the expected return at each state, based on the solution of a network-flow optimization derived from an interest function $\delta$.

A consequence of our work is to formalize why predicting value function-like objects is helpful in learning representations, as has been argued in the past (Sutton et al., 2011, 2016). We show how using these predictions as auxiliary tasks can be interpreted as a relaxation of our optimization problem. From our analysis, we hypothesize that auxiliary tasks that resemble adversarial value functions should give rise to good representations in practice. We complement our theoretical results with an empirical study in a simple grid world environment, focusing on the use of deep learning techniques to learn representations. We find that predicting adversarial value functions as auxiliary tasks leads to rich representations.

## 2 Setting

We consider an environment described by a Markov Decision Process $\langle \mathcal{X}, \mathcal{A}, r, P, \gamma \rangle$ (Puterman, 1994); $\mathcal{X}$ and $\mathcal{A}$ are finite state and action spaces, $P : \mathcal{X} \times \mathcal{A} \to \mathscr{P}(\mathcal{X})$ is the transition function, $\gamma$ the discount factor, and $r : \mathcal{X} \to \mathbb{R}$ the reward function. For a finite set $\mathcal{S}$, write $\mathscr{P}(\mathcal{S})$ for the probability simplex over $\mathcal{S}$. A (stationary) *policy* $\pi$ is a mapping $\mathcal{X} \to \mathscr{P}(\mathcal{A})$, also denoted $\pi(a \,|\, x)$. We denote the set of policies by $\mathcal{P} = \mathscr{P}(\mathcal{A})^{\mathcal{X}}$. We combine a policy $\pi$ with the transition function $P$ to obtain the state-to-state transition function $P^\pi(x' \,|\, x) := \sum_{a \in \mathcal{A}} \pi(a \,|\, x) P(x' \,|\, x, a)$. The *value function* $V^\pi$ describes the expected discounted sum of rewards obtained by following $\pi$:

$$V^\pi(x) = \mathbb{E}\Big[ \sum_{t=0}^{\infty} \gamma^t r(x_t) \,\big|\, x_0 = x, x_{t+1} \sim P^\pi(\cdot \,|\, x_t) \Big].$$

The value function satisfies Bellman's equation (Bellman, 1957): $V^\pi(x) = r(x) + \gamma \,\mathbb{E}_{P^\pi} V^\pi(x')$. Assuming there are $n = |\mathcal{X}|$ states, we view $r$ and $V^\pi$ as vectors in $\mathbb{R}^n$ and $P^\pi \in \mathbb{R}^{n \times n}$, such that

$$V^\pi = r + \gamma P^\pi V^\pi = (I - \gamma P^\pi)^{-1} r.$$

A *d-dimensional representation* is a mapping $\phi : \mathcal{X} \to \mathbb{R}^d$; $\phi(x)$ is the *feature vector* for state $x$. We write $\Phi \in \mathbb{R}^{n \times d}$ to denote the matrix whose rows are $\phi(\mathcal{X})$, and with some abuse of notation denote the set of $d$-dimensional representations by $\mathscr{R} \equiv \mathbb{R}^{n \times d}$. For a given representation and weight vector $\theta \in \mathbb{R}^d$, the linear approximation for a value function is

$$\hat{V}_{\phi,\theta}(x) := \phi(x)^\top \theta. \tag{1}$$

We consider the approximation minimizing the uniformly weighted squared error

$$\big\| \hat{V}_{\phi,\theta} - V^\pi \big\|_2^2 = \sum_{x \in \mathcal{X}} (\phi(x)^\top \theta - V^\pi(x))^2.$$

We denote by $\hat{V}_\phi^\pi$ the projection of $V^\pi$ onto the linear subspace $H = \big\{ \Phi\theta : \theta \in \mathbb{R}^d \big\}$.

### 2.1 Two-Part Networks

Most deep networks used in value-based reinforcement learning can be modelled as two interacting parts $\phi$ and $\theta$ which give rise to a linear approximation (Figure 1, left). Here, the representation $\phi$ can also be adjusted and is almost always nonlinear in $x$. Two-part networks are a simple framework in which to study the behaviour of representation learning in deep reinforcement learning. We will especially consider the use of $\phi(x)$ to make additional predictions, called *auxiliary tasks* following common usage, and whose purpose is to improve or stabilize the representation.

We study two-part networks in an idealized setting where the length $d$ of $\phi(x)$ is fixed and smaller than $n$, but the mapping is otherwise unconstrained. Even this idealized design offers interesting

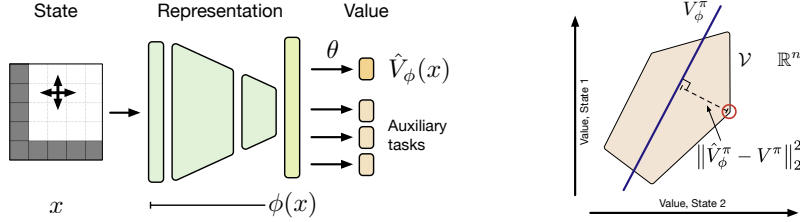

Figure 1: **Left.** A deep network viewed as a composition of two parts, one linear and one not. **Right.** The optimal representation $\phi^*$ is a linear subspace that cuts through the value polytope.

problems to study. We might be interested in sharing a representation across problems, as is often done in transfer or continual learning. In this context, auxiliary tasks may inform how the value function should generalize to these new problems. In many problems of interest, the weights $\theta$ can also be optimized more efficiently than the representation itself, warranting the view that the representation should be adapted using a different process (Levine et al., 2017; Chung et al., 2019).

Note that a trivial "value-as-feature" representation exists for the single-policy optimization problem

$$\text{minimize} \quad \left\| \hat{V}_\phi^\pi - V^\pi \right\|_2^2 \quad \text{w.r.t. } \phi \in \mathscr{R};$$

this approximation sets $\phi(x) = V^\pi(x)$ and $\theta = 1$. In this paper we take the stance that this is not a satisfying representation, and that a good representation should be in the service of a broader goal (e.g. control, transfer, or fairness).

## 3 Representation Learning by Approximating Value Functions

We measure the quality of a representation $\phi$ in terms of how well it can approximate all possible value functions, formalized as the *representation error*

$$L(\phi) := \max_{\pi \in \mathcal{P}} L(\phi; \pi), \quad L(\phi; \pi) := \left\| \hat{V}_\phi^\pi - V^\pi \right\|_2^2.$$

We consider the problem of finding the representation $\phi \in \mathscr{R}$ minimizing $L(\phi)$:

$$\text{minimize} \quad \max_{\pi \in \mathcal{P}} \left\| \hat{V}_\phi^\pi - V^\pi \right\|_2^2 \quad \text{w.r.t. } \phi \in \mathscr{R}. \tag{2}$$

In the context of our work, we call this the *representation learning problem* (RLP) and say that a representation $\phi^*$ is *optimal* when it minimizes the error in (2). Note that $L(\phi)$ (and hence $\phi^*$) depends on characteristics of the environment, in particular on both reward and transition functions.

We consider the RLP from a geometric perspective (Figure 1, right). Dadashi et al. (2019) showed that the set of value functions achieved by the set of policies $\mathcal{P}$, denoted

$$\mathcal{V} := \{V^\pi \in \mathbb{R}^n : \pi \in \mathcal{P}\},$$

forms a (possibly nonconvex) polytope. As previously noted, a representation $\phi$ defines a subspace $H$ of possible value approximations. The maximal error is achieved by the value function in $\mathcal{V}$ which is furthest along the subspace normal to $H$, since $\hat{V}_\phi^\pi$ is the orthogonal projection of $V^\pi$.

We say that $V \in \mathcal{V}$ is an *extremal vertex* if it is a vertex of the convex hull of $\mathcal{V}$. We will make use of the relationship between directions $\delta \in \mathbb{R}^d$, the set of extremal vertices, and the set of deterministic policies. The following lemma, based on a well-known notion of duality from convex analysis (Boyd and Vandenberghe, 2004), states this relationship formally.

**Lemma 1.** *Let* $\delta \in \mathbb{R}^n$ *and define the functional* $f_\delta(V) := \delta^\top V$, *with domain* $\mathcal{V}$. *Then* $f_\delta$ *is maximized by an extremal vertex* $U \in \mathcal{V}$, *and there is a deterministic policy* $\pi$ *for which* $V^\pi = U$. *Furthermore, the set of directions* $\delta \in \mathbb{R}^n$ *for which the maximum of* $f_\delta$ *is achieved by multiple extremal vertices has Lebesgue measure zero in* $\mathbb{R}^n$.

Denote by $\mathcal{P}_v$ the set of policies corresponding to extremal vertices of $\mathcal{V}$. We next derive an equivalence between the RLP and an optimization problem which only considers policies in $\mathcal{P}_v$.

**Theorem 1.** *For any representation $\phi \in \mathscr{R}$, the maximal approximation error measured over all value functions is the same as the error measured over the set of extremal vertices:*

$$\max_{\pi \in \mathcal{P}} \left\| \hat{V}_\phi^\pi - V^\pi \right\|_2^2 = \max_{\pi \in \mathcal{P}_v} \left\| \hat{V}_\phi^\pi - V^\pi \right\|_2^2.$$

Theorem 1 indicates that we can find an optimal representation by considering a finite (albeit exponential) number of value functions: Each extremal vertex corresponds to the value function of some deterministic policy, of which there are at most an exponential number. We will call these *adversarial value functions* (AVFs), because of the minimax flavour of the RLP.

Solving the RLP allows us to provide quantifiable guarantees on the performance of certain value-based learning algorithms. For example, in the context of least-squares policy iteration (LSPI; Lagoudakis and Parr, 2003), minimizing the representation error $L$ directly improves the performance bound. By contrast, we cannot have the same guarantee if $\phi$ is learned by minimizing the approximation error for a single value function.

**Corollary 1.** *Let $\phi^*$ be an optimal representation in the RLP. Consider the sequence of policies $\pi_0, \pi_1, \ldots$ derived from LSPI using $\phi^*$ to approximate $V^{\pi_0}, V^{\pi_1}, \ldots$ under a uniform sampling of the state-space. Then there exists an MDP-dependent constant $C \in \mathbb{R}$ such that*

$$\limsup_{k \to \infty} \left\| V^* - V^{\pi_k} \right\|_2^2 \leq CL(\phi^*).$$

This result is a direct application of the quadratic norm bounds given by Munos (2003), in whose work the constant is made explicit. We emphasize that the result is illustrative; our approach should enable similar guarantees in other contexts (e.g. Munos, 2007; Petrik and Zilberstein, 2011).

### 3.1 The Structure of Adversarial Value Functions

The RLP suggests that an agent trained to predict various value functions should develop a good state representation. Intuitively, one may worry that there are simply too many "uninteresting" policies, and that a representation learned from their value functions emphasizes the wrong quantities. However, the search for an optimal representation $\phi^*$ is closely tied to the much smaller set of adversarial value functions (AVFs). The aim of this section is to characterize the structure of AVFs and show that they form an *interesting* subset of all value functions. From this, we argue that their use as auxiliary tasks should also produce structured representations.

From Lemma 1, recall that an AVF is geometrically defined using a vector $\delta \in \mathbb{R}^n$ and the functional $f_\delta(V) := \delta^\top V$, which the AVF maximizes. Since $f_\delta$ is restricted to the value polytope, we can consider the equivalent policy-space functional $g_\delta : \pi \mapsto \delta^\top V^\pi$. Observe that

$$\max_{\pi \in \mathcal{P}} g_\delta(\pi) = \max_{\pi \in \mathcal{P}} \delta^\top V^\pi = \max_{\pi \in \mathcal{P}} \sum_{x \in \mathcal{X}} \delta(x) V^\pi(x). \tag{3}$$

In this optimization problem, the vector $\delta$ defines a weighting over the state space $\mathcal{X}$; for this reason, we call $\delta$ an *interest function* in the context of AVFs. Whenever $\delta \geq 0$ componentwise, we recover the optimal value function, irrespective of the exact magnitude of $\delta$ (Bertsekas, 2012). If $\delta(x) < 0$ for some $x$, however, the maximization becomes a minimization. As the next result shows, the policy maximizing $f_\delta(\pi)$ depends on a network flow $d_\pi$ derived from $\delta$ and the transition function $P$.

**Theorem 2.** *Maximizing the functional $g_\delta$ is equivalent to finding a network flow $d_\pi$ that satisfies a reverse Bellman equation:*

$$\max_{\pi \in \mathcal{P}} \delta^\top V^\pi = \max_{\pi \in \mathcal{P}} d_\pi^\top r, \qquad d_\pi = \delta + \gamma P^{\pi\top} d_\pi.$$

*For a policy $\tilde{\pi}$ maximizing the above we have*

$$V^{\tilde{\pi}}(x) = r(x) + \gamma \left\{ \begin{array}{ll} \max_{a \in \mathcal{A}} \mathbb{E}_{x' \sim P} V^{\tilde{\pi}}(x') & d_{\tilde{\pi}}(x) > 0, \\ \min_{a \in \mathcal{A}} \mathbb{E}_{x' \sim P} V^{\tilde{\pi}}(x') & d_{\tilde{\pi}}(x) < 0. \end{array} \right.$$

**Corollary 2.** *There are at most $2^n$ distinct adversarial value functions.*

The vector $d_\pi$ corresponds to the sum of discounted interest weights flowing through a state $x$, similar to the dual variables in the theory of linear programming for MDPs (Puterman, 1994). Theorem 2, by way of the corollary, implies that there are fewer AVFs ($\leq 2^n$) than deterministic policies ($= |\mathcal{A}|^n$). It also implies that AVFs relate to a reward-driven purpose, similar to how the optimal value function describes the goal of maximizing return. We will illustrate this point empirically in Section 4.1.

## 3.2 Relationship to Auxiliary Tasks

So far we have argued that solving the RLP leads to a representation which is optimal in a meaningful sense. However, solving the RLP seems computationally intractable: there are an exponential number of deterministic policies to consider (Prop. 1 in the appendix gives a quadratic formulation with quadratic constraints). Using interest functions does not mitigate this difficulty: the computational problem of finding the AVF for a single interest function is NP-hard, even when restricted to deterministic MDPs (Prop. 2 in the appendix).

Instead, in this section we consider a relaxation of the RLP and show that this relaxation describes existing representation learning methods, in particular those that use auxiliary tasks. Let $\xi$ be some distribution over $\mathbb{R}^n$. We begin by replacing the maximum in (2) by an expectation:

$$\text{minimize} \quad \mathop{\mathbb{E}}_{V \sim \xi} \left\| \hat{V}_\phi - V \right\|_2^2 \quad \text{w.r.t. } \phi \in \mathscr{R}. \tag{4}$$

The use of the expectation offers three practical advantages over the use of the maximum. First, this leads to a differentiable objective which can be minimized using deep learning techniques. Second, the choice of $\xi$ gives us an additional degree of freedom; in particular, $\xi$ needs not be restricted to the value polytope. Third, the minimizer in (4) is easily characterized, as the following theorem shows.

**Theorem 3.** *Let $u_1^*, \ldots, u_d^* \in \mathbb{R}^n$ be the principal components of the distribution $\xi$, in the sense that*

$$u_i^* := \mathop{\arg\max}_{u \in B_i} \mathop{\mathbb{E}}_{V \sim \xi} (u^\top V)^2, \text{ where } B_i := \{u \in \mathbb{R}^n : \|u\|_2^2 = 1, u^\top u_j^* = 0 \ \forall j < i\}.$$

*Equivalently, $u_1^*, \ldots, u_d^*$ are the eigenvectors of $\mathbb{E}_\xi VV^\top \in \mathbb{R}^{n \times n}$ with the $d$ largest eigenvalues. Then the matrix $[u_1^*, \ldots, u_d^*] \in \mathbb{R}^{n \times d}$, viewed as a map $\mathcal{X} \to \mathbb{R}^d$, is a solution to (4). When the principal components are uniquely defined, any minimizer of (4) spans the same subspace as $u_1^*, \ldots, u_d^*$.*

One may expect the quality of the learned representation to depend on how closely the distribution $\xi$ relates to the RLP. From an auxiliary tasks perspective, this corresponds to choosing tasks that are in some sense useful. For example, generating value functions from the uniform distribution over the set of policies $\mathcal{P}$, while a natural choice, may put too much weight on "uninteresting" value functions.

In practice, we may further restrict $\xi$ to a finite set $\boldsymbol{V}$. Under a uniform weighting, this leads to a *representation loss*

$$L(\phi; \boldsymbol{V}) := \sum_{V \in \boldsymbol{V}} \left\| \hat{V}_\phi - V \right\|_2^2 \tag{5}$$

which corresponds to the typical formulation of an auxiliary-task loss (e.g. Jaderberg et al., 2017). In a deep reinforcement learning setting, one typically minimizes (5) using stochastic gradient descent methods, which scale better than batch methods such as singular value decomposition (but see Wu et al. (2019) for further discussion).

Our analysis leads us to conclude that, in many cases of interest, the use of auxiliary tasks produces representations that are close to the principal components of the set of tasks under consideration. If $\boldsymbol{V}$ is well-aligned with the RLP, minimizing $L(\phi; \boldsymbol{V})$ should give rise to a reasonable representation. To demonstrate the power of this approach, in Section 4 we will study the case when the set $\boldsymbol{V}$ is constructed by sampling AVFs – emphasizing the policies that support the solution to the RLP.

## 3.3 Relationship to Proto-Value Functions

Proto-value functions (Mahadevan and Maggioni, 2007, PVF) are a family of representations which vary smoothly across the state space. Although the original formulation defines this representation as the largest-eigenvalue eigenvectors of the Laplacian of the transition function's graphical structure, recent formulations use the top singular vectors of $(I - \gamma P^\pi)^{-1}$, where $\pi$ is the uniformly random policy (Stachenfeld et al., 2014; Machado et al., 2017; Behzadian and Petrik, 2018).

In line with the analysis of the previous section, proto-value functions can also be interpreted as defining a set of value-based auxiliary tasks. Specifically, if we define an indicator reward function $r_y(x) := \mathbb{I}_{[x=y]}$ and a set of value functions $\boldsymbol{V} = \{(I - \gamma P^\pi)^{-1} r_y\}_{y \in \mathcal{X}}$ with $\pi$ the uniformly random policy, then any $d$-dimensional representation that minimizes (5) spans the same basis as the $d$-dimensional PVF (up to the bias term). This suggests a connection with hindsight experience replay (Andrychowicz et al., 2017), whose auxiliary tasks consists in reaching previously experienced states.

# 4 Empirical Studies

In this section we complement our theoretical analysis with an experimental study. In turn, we take a closer look at 1) the **structure** of adversarial value functions, 2) the **shape of representations** learned using AVFs, and 3) the **performance profile** of these representations in a control setting. Our eventual goal is to demonstrate that the RLP, which is based on approximating value functions, gives rise to representations that are both interesting and comparable to previously proposed schemes. Our concrete instantiation (Algorithm 1) uses the representation loss (5). As-is, this algorithm is of limited practical relevance (our AVFs are learned using a tabular representation) but we believe provides an inspirational basis for further developments.

---

**Algorithm 1** Representation learning using AVFs

---

**input** $k$ – desired number of AVFs, $d$ – desired number of features.
  Sample $\delta_1, \ldots, \delta_k \sim [-1, 1]^n$
  Compute $\mu_i = \arg\max_\pi \delta_i^\top V^\pi$ using a policy gradient method
  Find $\phi^* = \arg\min_\phi L(\phi; \{V^{\mu_1}, \ldots, V^{\mu_k}\})$ (Equation 5)

---

We perform all of our experiments within the four-room domain (Sutton et al., 1999; Solway et al., 2014; Machado et al., 2017, Figure 2, see also Appendix H.1).

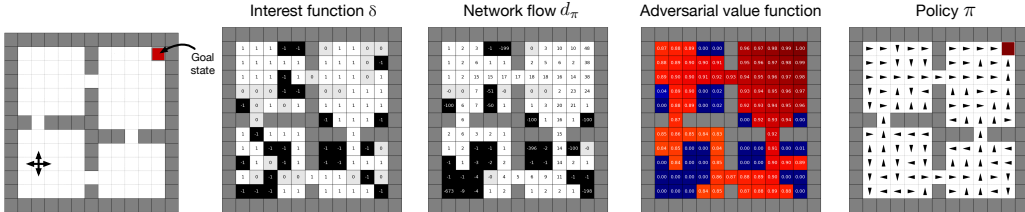

Figure 2: **Leftmost.** The four-room domain. **Other panels.** An interest function $\delta$, the network flow $d_\pi$, the corresponding adversarial value function (blue/red = low/high value) and its policy.

We consider a two-part network where we pretrain $\phi$ end-to-end to predict a set of value functions. Our aim here is to compare the effects of using different sets of value functions, including AVFs, on the learned representation. As our focus is on the efficient use of a $d$-dimensional representation (with $d < n$, the number of states), we encode individual states as one-hot vectors and map them into $\phi(x)$ without capacity constraints. Additional details may be found in Appendix H.

## 4.1 Adversarial Value Functions

Our first set of results studies the structure of adversarial value functions in the four-room domain. We generated interest functions by assigning a value $\delta(x) \in \{-1, 0, 1\}$ uniformly at random to each state $x$ (Figure 2, left). We restricted $\delta$ to these discrete choices for illustrative purposes.

We then used model-based policy gradient (Sutton et al., 2000) to find the policy maximizing $\sum_{x \in \mathcal{X}} \delta(x) V^\pi(x)$. We observed some local minima or accumulation points but as a whole reasonable solutions were found. The resulting network flow and AVF for a particular sample are shown in Figure 2. For most states, the signs of $\delta$ and $d_\pi$ agree; however, this is not true of all states (larger version and more examples in appendix, Figures 6, 7). As expected, states for which $d_\pi > 0$ (respectively, $d_\pi < 0$) correspond to states maximizing (resp. minimizing) the value function. Finally, we remark on the "flow" nature of $d_\pi$: trajectories over minimizing states accumulate in corners or loops, while those over maximizing states flow to the goal. We conclude that AVFs exhibit interesting structure, and are generated by policies that are not random (Figure 2, right). As we will see next, this is a key differentiator in making AVFs good auxiliary tasks.

## 4.2 Representation Learning with AVFs

We next consider the representations that arise from training a deep network to predict AVFs (denoted AVF from here on). We sample $k = 1000$ interest functions and use Algorithm 1 to generate $k$ AVFs.

We combine these AVFs into the representation loss (5) and adapt the parameters of the deep network using Rmsprop (Tieleman and Hinton, 2012).

We contrast the AVF-driven representation with one learned by predicting the value function of random deterministic policies (RP). Specifically, these policies are generated by assigning an action uniformly at random to each state. We also consider the value function of the uniformly random policy (VALUE). While we make these choices here for concreteness, other experiments yielded similar results (e.g. predicting the value of the optimal policy; appendix, Figure 8). In all cases, we learn a $d = 16$ dimensional representation, not including the bias unit.

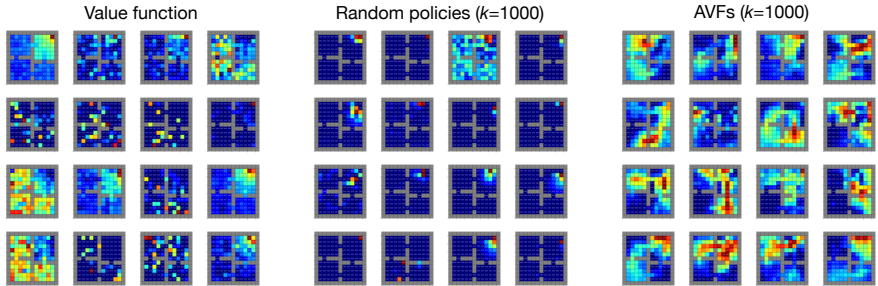

Figure 3: 16-dimensional representations learned by predicting a single value function, the value functions of 1000 random policies, or 1000 AVFs sampled using Algorithm 1. Each panel element depicts the activation of a given feature across states, with blue/red indicating low/high activation.

Figure 3 shows the representations learned by the three methods. The features learned by VALUE resemble the value function itself (top left feature) or its negated image (bottom left feature). Coarsely speaking, these features capture the general distance to the goal but little else. The features learned by RP are of even worse quality. This is because almost all random deterministic policies cause the agent to avoid the goal (appendix, Figure 12). The representation learned by AVF, on the other hand, captures the structure of the domain, including paths between distal states and focal points corresponding to rooms or parts of rooms.

Although our focus is on the use of AVFs as auxiliary tasks to a deep network, we observe the same results when discovering a representation using singular value decomposition (Section 3.2), as described in Appendix I. All in all, our results illustrate that, among all value functions, AVFs are particularly useful auxiliary tasks for representation learning.

## 4.3 Learning the Optimal Policy

In a final set of experiments, we consider learning a reward-maximizing policy using a pretrained representation and a model-based version of the SARSA algorithm (Rummery and Niranjan, 1994; Sutton and Barto, 1998). We compare the value-based and AVF-based representations from the previous section (VALUE and AVF), and also proto-value functions (PVF; details in Appendix H.3).

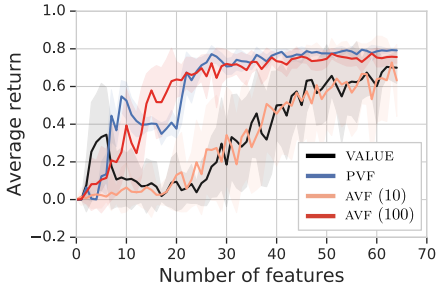

Figure 4: Average discounted return achieved by policies learned using a representation produced by VALUE, AVF, or PVF. Average is over 20 random seeds and shading gives standard deviation.

We report the quality of the learned policies after training, as a function of $d$, the size of the representation. Our quality measure is the average return from the designated start state (bottom left). Results are provided in Figure 4 and Figure 13 (appendix). We observe a failure of the VALUE representation to provide a useful basis for learning a good policy, even as $d$ increases; while the representation is not rank-deficient, the features do not help reduce the approximation error.

In comparison, our AVF representations perform similarly to PVFs. Increasing the number of auxiliary tasks also leads to better representations; recall that PVF implicitly uses $n = 104$ auxiliary tasks.

## 5   Related Work

Our work takes inspiration from research in basis or feature construction for reinforcement learning. Ratitch and Precup (2004), Foster and Dayan (2002), Menache et al. (2005), Yu and Bertsekas (2009), Bhatnagar et al. (2013), and Song et al. (2016) consider methods for adapting parametrized basis functions using iterative schemes. Including Mahadevan and Maggioni (2007)'s proto-value functions, a number of works (we note Dayan, 1993; Petrik, 2007; Mahadevan and Liu, 2010; Ruan et al., 2015; Barreto et al., 2017) have used characteristics of the transition structure of the MDP to generate representations; these are the closest in spirit to our approach, although none use the reward or consider the geometry of the space of value functions. Parr et al. (2007) proposed constructing a representation from successive Bellman errors, Keller et al. (2006) used dimensionality reduction methods; finally Hutter (2009) proposes a universal scheme for selecting representations.

Deep reinforcement learning algorithms have made extensive use of auxiliary tasks to improve agent performance, beginning perhaps with universal value function approximators (Schaul et al., 2015) and the UNREAL architecture (Jaderberg et al., 2017); see also Dosovitskiy and Koltun (2017), François-Lavet et al. (2018) and, more tangentially, van den Oord et al. (2018). Levine et al. (2017) and Chung et al. (2019) make explicit use of two-part network to derive more sample efficient deep reinforcement learning algorithms. Veeriah et al. (2019) use a meta-gradient approach to generate auxiliary tasks. The notion of augmenting an agent with side predictions is not new, with roots in TD models (Sutton, 1995), predictive state representations (Littman et al., 2002), and the Horde architecture (Sutton et al., 2011), itself inspired by the work of Selfridge (1959).

A number of works quantify or explain the usefulness of a representation. Parr et al. (2008) demonstrated that a good representation should support a good approximation of both reward and expected next state. We conjecture that the relaxed problem (4) trades these two quantities off in a principled fashion. Li et al. (2006); Abel et al. (2016) consider the approximation error that arises from state abstraction. More recently, Nachum et al. (2019) provide some interesting guarantees in the context of hierarchical reinforcement learning, while Such et al. (2019) visualizes the representations learned by Atari-playing agents. Finally, Bertsekas (2018) remarks on the two-part network we study here.

## 6   Conclusion

In this paper we studied the notion of an adversarial value function, derived from a geometric perspective on representation learning in RL. Our work shows that adversarial value functions exhibit interesting structure, and are good auxiliary tasks when learning a representation of an environment. We believe our work to be the first to provide formal evidence as to the usefulness of predicting value functions for shaping an agent's representation.

Our work opens up the possibility of automatically generating auxiliary tasks in deep reinforcement learning, analogous to how deep learning itself enabled a move away from hand-crafted features. To do so, we expect that a number of practical challenges will need to be overcome:

**Off-policy learning.** A practical implementation will require learning AVFs concurrently with the main task. Doing so results in off-policy learning, whose negative effects are well-documented even in recent applications (e.g. van Hasselt et al., 2018).

**Policy parametrization.** AVFs are the value function of deterministic policies. While a natural choice is to look for policies that maximize representation error, this poses the problem of how to parametrize the policies themselves. In particular, a policy parametrized using the representation $\phi$ may not provide a sufficient degree of "adversariality".

**Smoothness in the interest function.** In continuous or large state spaces, it is desirable for interest functions to incorporate some degree of smoothness, rather than vary rapidly from state to state. It is not clear how to control this smoothness in a principled manner.

From a mathematical perspective, our formulation of the RLP was made with both convenience and geometry in mind. Conceptually, it may be interesting to consider our approach in other norms,

including the weighted norms used in approximation results. Practically, this would translate into an emphasis on "interesting" value functions, for example by giving additional weight to the optimal value function and its neighbouring AVFs.

## 7   Acknowledgements

The authors thank the many people who helped shape this project through discussions and feedback on early and late drafts: Lihong Li, George Tucker, Doina Precup, Ofir Nachum, Csaba Szepesvári, Georg Ostrovski, Marek Petrik, Marlos Machado, Tim Lillicrap, Danny Tarlow, Hugo Larochelle, Saurabh Kumar, Carles Gelada, Rémi Munos, David Silver, and André Barreto. Special thanks also to Philip Thomas and Scott Niekum, who gave this project its initial impetus.

## 8   Author Contributions

M.G.B., W.D., D.S., and N.L.R. conceptualized the representation learning problem. M.G.B., W.D., T.L., A.A.T., R.D., D.S., and N.L.R. contributed to the theoretical results. M.G.B., W.D., P.S.C., R.D., and C.L. performed experiments and collated results. All authors contributed to the writing.

## Footnotes

[1]Google Research [2]DeepMind [3]Mila, Université de Montréal [4]University of Alberta [5]University of Oxford

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
