[Supplementary Material]

# A    Proof of Lemma 1

Consider the value polytope $\mathcal{V}$. We have using Corollary 1 of Dadashi et al. (2019) that

$$\mathcal{V} \subseteq \text{CONV}(\mathcal{V}) = \text{CONV}(V^{\pi_1}, \dots, V^{\pi_m}), \tag{6}$$

where $\pi_1, \dots, \pi_m$ is a finite collection of deterministic policies. We assume that this set of policies is of minimal cardinality e.g. the value functions $V^{\pi_1}, \dots, V^{\pi_m}$ are distinct.

The optimization problem $\max_{V \in \mathcal{V}} \delta^\top V$ is equivalent to the linear program $\max_{V \in \text{CONV}(\mathcal{V})} \delta^\top V$, and the maximum is reached at a vertex $U$ of the convex hull of $\mathcal{V}$ (Boyd and Vandenberghe, 2004). By (6), $U$ is the value function of a deterministic policy. Now consider $\delta \in \mathbb{R}^n$ such that $f_\delta$ attains its maximum over multiple elements of the convex hull. By hypothesis, there must be two policies $\pi_i$, $\pi_j$ such that $V^{\pi_i} \neq V^{\pi_j}$ and

$$\max_{V \in \mathcal{V}} \delta^\top V = \delta^\top V^{\pi_i} = \delta^\top V^{\pi_j},$$

and thus

$$\delta^\top (V^{\pi_i} - V^{\pi_j}) = 0. \tag{7}$$

Write $\Delta$ for the ensemble of such $\delta$. We have from (7):

$$\Delta \subseteq \bigcup_{1 \leq i < j \leq m} \{\delta \in \mathbb{R}^n \mid \delta^T (V^{\pi_i} - V^{\pi_j}) = 0\}.$$

As $V^{\pi_1}, \dots, V^{\pi_m}$ are distinct, $\Delta$ is included in a finite union of hyperplanes (recall that hyperplanes of $\mathbb{R}^n$ are vector spaces of dimension $n - 1$). The Lebesgue measure of a hyperplane is 0 (in $\mathbb{R}^n$), hence a finite union of hyperplanes also has Lebesgue measure 0. Hence $\Delta$ itself has Lebesgue measure of 0 in $\mathbb{R}^n$.

# B    Proof of Corollary 2

Similarly to the proof of Lemma 1, we introduce $V^{\pi_1}, \dots, V^{\pi_m}$ which are the distinct vertices of the convex hull of the value polytope $\mathcal{V}$. Note that $\pi_1, \dots, \pi_m$ are deterministic policies. We shall show that there are at most $2^n$ such vertices.

Recall the definition of a cone in $\mathbb{R}^n$: $C$ is a cone in $\mathbb{R}^n$ if $\forall v \in C, \forall \alpha \geq 0, \alpha v \in C$. For each vertex $V^{\pi_i}$, Rockafellar and Wets (2009, Theorem 6.12) states that there is an associated cone $C_i$ of nonzero Lebesgue measure in $\mathbb{R}^n$ such that

$$\forall \delta \in C_i, \, \arg\max_{V \in \mathcal{V}} \delta^\top V = V^{\pi_i}.$$

Now using Theorem 2, we have

$$\max_{V \in \mathcal{V}} \delta^\top V = \max_{\pi \in \mathcal{P}} d_\pi^\top r, \text{ where } d_\pi = (I - \gamma P^{\pi\top})^{-1}\delta.$$

For all $\delta \in C_i$ the corresponding policy $\pi_i$ is the same (by hypothesis). For such a $\delta$, define $d_{\pi_i, \delta} := (I - \gamma P^{\pi_i \top})^{-1}\delta$, such that

$$\delta^\top V^{\pi_i} = d_{\pi_i, \delta}^\top r.$$

Because $C_i$ is a cone of nonzero Lebesgue measure in $\mathbb{R}^n$, we have $span(C_i) = \mathbb{R}^n$. Combined with the fact that $(I - \gamma P^{\pi_i \top})^{-1}$ is full rank, this implies we can find a direction $\delta_i$ in $C_i$ for which $d_{\pi_i, \delta_i}(x) \neq 0$ for all $x \in \mathcal{X}$. For this $\delta_i$, using Theorem 2 we have:

$$V^{\pi_i}(x) = r(x) + \gamma \begin{cases} \max_{a \in \mathcal{A}} \mathbb{E}_{x' \sim P} V^{\pi_i}(x') & d_{\pi_i, \delta_i}(x) > 0, \\ \min_{a \in \mathcal{A}} \mathbb{E}_{x' \sim P} V^{\pi_i}(x') & d_{\pi_i, \delta_i}(x) < 0, \end{cases} \tag{8}$$

and each state is "strictly" a maximizer or minimizer (the purpose of our cone argument was to avoid the undefined case where $d_{\pi_i, \delta_i}(x) = 0$). Now define $\sigma_i \in \{-1, 1\}^n$, $\sigma_i(x) = \text{sign}(d_{\pi_i, \delta_i}(x))$. We have:

$$V^{\pi_i}(x) = r(x) + \gamma \sigma_i(x) \max_{a \in \mathcal{A}} \sigma_i(x) \mathbb{E}_{x' \sim P} V^{\pi_i}(x')$$
$$= \mathcal{T}_{\sigma_i} V^{\pi_i}(x)$$

where $\mathcal{T}_\sigma V(x) = r(x) + \gamma\sigma(x)\max_{a\in\mathcal{A}}\sigma(x)\mathbb{E}_{x'\sim P}V(x')$ for $\sigma \in \{-1,1\}^n$. We show that $\mathcal{T}_\sigma$ is a contraction mapping: for any $x \in \mathcal{X}$ and $\sigma \in \{-1,1\}^n$,

$$
\begin{aligned}
|\mathcal{T}_\sigma V_1(x) - \mathcal{T}_\sigma V_2(x)| &= |r(x) + \gamma\sigma(x)\max_{a\in\mathcal{A}}\sigma(x)\mathbb{E}_{x'\sim P}V_1(x') - r(x) - \gamma\sigma(x)\max_{a\in\mathcal{A}}\sigma(x)\mathbb{E}_{x'\sim P}V_2(x')| \\
&= \gamma|\max_{a\in\mathcal{A}}\sigma(x)\mathbb{E}_{x'\sim P}V_1(x') - \max_{a\in\mathcal{A}}\sigma(x)\mathbb{E}_{x'\sim P}V_2(x')| \\
&\le \gamma\max_{a\in\mathcal{A}}|\sigma(x)\mathbb{E}_{x'\sim P}V_1(x') - \sigma(x)\mathbb{E}_{x'\sim P}V_2(x')| \\
&\le \gamma\max_{a\in\mathcal{A}}\max_{x'\in\mathcal{X}}|V_1(x') - V_2(x')| \\
&= \gamma\max_{x'\in\mathcal{X}}|V_1(x') - V_2(x')|.
\end{aligned}
$$

Therefore, $\|\mathcal{T}_\sigma V_1 - \mathcal{T}_\sigma V_2\|_\infty \le \gamma\|V_1 - V_2\|_\infty$ and $\mathcal{T}_\sigma$ is a $\gamma$-contraction in the supremum norm. By Banach's fixed point theorem $V^{\pi_i}$ is its unique fixed point.

We showed that each vertex $V^{\pi_i}$ of the value function polytope $\mathcal{V}$ is the fixed point of an operator $\mathcal{T}_{\sigma_i}$. Since there are $2^n$ such operators, there are at most $2^n$ vertices.

## C Proof of Theorem 1

We will show that the maximization over $\mathcal{P}$ is the same as the maximization over $\mathcal{P}_v$.

Let $\Pi_\phi$ be the projection matrix onto the hyperplane $H$ spanned by the basis induced by $\phi$. We write

$$
\begin{aligned}
\|\hat{V}^\pi_\phi - V^\pi\|_2^2 &= \|\Pi_\phi V^\pi - V^\pi\|_2^2 \\
&= \|(\Pi_\phi - I)V^\pi\|_2^2 \\
&= V^{\pi\top}(\Pi_\phi - I)^\top(\Pi_\phi - I)V^\pi \\
&= V^{\pi\top}(I - \Pi_\phi)V^\pi
\end{aligned}
$$

because $\Pi_\phi$ is idempotent ($\Pi_\phi^2 = \Pi_\phi$). The eigenvalues of $A = I - \Pi_\phi$ are 1 and 0, and the eigenvectors corresponding to eigenvalue 1 are normal to $H$. Because we are otherwise free to choose any basis spanning the subspace normal to $H$, there is a unit vector $\delta$ normal to $H$ for which

$$
\begin{aligned}
\max_\pi \|\hat{V}^\pi_\phi - V^\pi\|_2^2 &= \max_{V^\pi\in\mathcal{V}}\|\hat{V}^\pi_\phi - V^\pi\|_2^2 \\
&= \max_{V^\pi\in\mathcal{V}}V^{\pi\top}\delta\delta^\top V^\pi.
\end{aligned}
$$

Denote the value function maximizing this quantity by $V^\pi_{\text{MAX}}$. This $\delta$ can be chosen so that $\delta^\top V^\pi_{\text{MAX}} > 0$ (if not, take $\delta' = -\delta$). Then $V^\pi_{\text{MAX}}$ is also the maximizer of $f(V) := \delta^\top V$ over $\mathcal{V}$, and Lemma 1 tells us that $V^\pi_{\text{MAX}}$ is an extremal vertex.

## D Proof of Theorem 2

To begin, note that

$$
\begin{aligned}
\delta^\top V^\pi &= \delta^\top (I - \gamma P^\pi)^{-1}r \\
&= (I - \gamma P^{\pi\top})^{-1}\delta^\top r \\
&= d_\pi^\top r,
\end{aligned}
$$

as required.

Now, we choose an indexing for states in $\mathcal{S}$ and will refer to states by their index.

Let $\pi$ be the policy maximizing $\delta^\top V^\pi$ and consider some $x^* \in \mathcal{S}$. We assume without loss of generality that $x^*$ is the first state in the previous ordering. Recall that $n = |\mathcal{S}|$.

The theorem states that policy $\pi$ chooses the highest-valued action at $x^*$ if $d_\pi(x^*) > 0$, and the lowest-valued action if $d_\pi(x^*) < 0$. Writing $P^\pi_{x^*} := P^\pi(\cdot\,|\,x^*)$ for conciseness, this is equivalent to

$$
r(x^*) + \mathbb{E}_{x'\sim P^\pi_{x^*}}V^\pi(x') = \max_{\pi'}r(x^*) + \mathbb{E}_{x'\sim P^{\pi'}_{x^*}}V^\pi(x'),
$$

for $d_\pi(x^*) > 0$, and conversely with a $\min_{\pi'}$ for $d_\pi(x^*) < 0$ (equivalently, $\mathcal{T}^\pi V^\pi(x^*) \geq \mathcal{T}^{\pi'} V^\pi(x^*)$ for all $\pi' \in \mathcal{P}$ or $\mathcal{T}^\pi V^\pi(x^*) \leq \mathcal{T}^{\pi'} V^\pi(x^*)$ in operator notation).

We write the transition matrix $P^\pi$ as follows

$$P^\pi = \begin{pmatrix} L_1^\pi \\ \vdots \\ L_n^\pi \end{pmatrix}.$$

Where $L_i^\pi = \big(P^\pi(x_1 \mid x_i), \cdots, P^\pi(x_n \mid x_i)\big)$ is $P^\pi$'s $i$-th row.

Then we express the transition matrix as $P^\pi = A^\pi + B^\pi$, with $A^\pi$ and $B^\pi$ given by

$$A^\pi = \begin{pmatrix} 0 \\ L_2^\pi \\ \vdots \\ L_n^\pi \end{pmatrix} \qquad B^\pi = \begin{pmatrix} L_1^\pi \\ 0 \\ \vdots \\ 0 \end{pmatrix}.$$

We can then write

$$
\begin{aligned}
V^\pi &= r + \gamma P^\pi V^\pi \\
&= r + \gamma(A^\pi + B^\pi)V^\pi \\
\Rightarrow\ V^\pi &= (I - \gamma A^\pi)^{-1}(r + \gamma B^\pi V^\pi).
\end{aligned}
$$

This is an application of matrix splitting (e.g Puterman, 1994). The invertibility of $(I - \gamma A^\pi)$ is guaranteed because $A^\pi$ is a substochastic matrix. The first term of the r.h.s corresponds to the expected sum of discounted rewards when following $\pi$ until reaching $x^*$, while the second term is the expected sum of discounted rewards received after leaving from $x^*$ and following policy $\pi$.

Note that $(I - \gamma A^\pi)^{-1}$ does not depend on $\pi(\cdot \mid x^*)$ and that

$$B^\pi V^\pi = \begin{pmatrix} \mathbb{E}_{x' \sim P_{x^*}^\pi} V^\pi(x') \\ 0 \\ \vdots \\ 0 \end{pmatrix}.$$

Write $C^\pi = (I - \gamma A^{\pi\top})^{-1}\delta$. We have

$$
\begin{aligned}
\delta^\top V^\pi &= \delta^\top (I - \gamma A^\pi)^{-1}(r + \gamma B^\pi V^\pi) \\
&= C^{\pi\top}(r + \gamma B^\pi V^\pi) \\
&= C^{\pi\top} r + C^\pi(x^*) \mathop{\mathbb{E}}_{x' \sim P_{x^*}^\pi} V^\pi(x').
\end{aligned}
$$

Now by assumption,

$$\delta^\top V^\pi \geq \delta^\top V^{\pi'} \tag{9}$$

for any other policy $\pi' \in \mathcal{P}$. Take $\pi'$ such that $\pi'(\cdot \mid x) = \pi(\cdot \mid x)$ everywhere but $x^*$; then $C^\pi = C^{\pi'}$ and (9) implies that

$$C^\pi(x^*) \mathop{\mathbb{E}}_{x' \sim P_{x^*}^\pi} V^\pi(x') \geq C^\pi(x^*) \mathop{\mathbb{E}}_{x' \sim P_{x^*}^{\pi'}} V^\pi(x').$$

Hence $\pi$ must pick the maximizing action in $x^*$ if $C^\pi(x^*) > 0$, and the minimizing action if $C^\pi(x^*) < 0$.

To conclude the proof, we show that $d_\pi(x^*)$ and $C^\pi(x^*)$ have the same sign. We write

$$d_\pi = \delta + \gamma(A^{\pi\top} + B^{\pi\top})d_\pi.$$

Then

$$(I - \gamma A^{\pi\top})d_\pi = \delta + \gamma B^{\pi\top}d_\pi$$
$$\Rightarrow \quad d_\pi = C^\pi + \gamma(I - \gamma A^{\pi\top})^{-1}B^{\pi\top}d_\pi$$
$$= \sum_{k=0}^{\infty}(\gamma(I - \gamma A^{\pi\top})^{-1}B^{\pi\top})^k C^\pi$$
$$= \sum_{k=0}^{\infty}\gamma^k(D^{\pi\top})^k C^\pi.$$

Where $D^\pi = B^\pi(I - \gamma A^\pi)^{-1}$ is a matrix with non-negative components. Because $B^\pi$ is sparse every row of $(D^\pi)^k$ is null except for the first one. We can write

$$(D^{\pi k})^\top = \begin{pmatrix} d_{11}^k \, 0 \cdots 0 \\ \vdots \\ d_{1n}^k \, 0 \cdots 0 \end{pmatrix} \quad \forall i, \, d_{1i}^k \geq 0.$$

And

$$d_\pi(x^*) = \Big(\sum_{k=0}^{\infty}\gamma^k d_{11}^k\Big) C^\pi(x^*).$$

Hence $C^\pi(x^*)$ and $d_\pi(x^*)$ have the same sign.

# E  Proof of Theorem 3

We first transform (4) in a equivalent problem. Let $V \in \mathbb{R}^n$, and denote by $\hat{V}_\phi := \Pi_\phi V$ the orthogonal projection of $V$ onto the subspace spanned by the columns of $\Phi$. From Pythagoras' theorem we have, for any $V \in \mathbb{R}^n$

$$\big\|V\big\|_2^2 = \big\|\hat{V}_\phi - V\big\|_2^2 + \big\|\hat{V}_\phi\big\|_2^2$$

Then

$$\min_{\phi\in\mathcal{R}}\mathop{\mathbb{E}}_{V\sim\xi}\big\|\hat{V}_\phi - V\big\|_2^2 = \min_{\phi\in\mathcal{R}}\mathop{\mathbb{E}}_{V\sim\xi}\big[\big\|V\big\|_2^2 - \big\|\Pi_\phi V\big\|_2^2\big]$$
$$= \max_{\phi\in\mathcal{R}}\mathop{\mathbb{E}}_{V\sim\xi}\big\|\Pi_\phi V\big\|_2^2.$$

Let $u_1^*, \ldots, u_d^*$ the principal components defined in Theorem 3. These form an orthonormal basis. Hence $u_1^*, \ldots, u_d^*$ is equivalently a solution of

$$\max_{\substack{u_1,\ldots,u_d\in\mathbb{R}^n \\ \text{orthonormal}}}\sum_{i=1}^{d}\mathop{\mathbb{E}}_{V\sim\xi}(u_i^\top V)_2^2 = \max_{\substack{u_1,\ldots,u_d\in\mathbb{R}^n \\ \text{orthonormal}}}\sum_{i=1}^{d}\mathop{\mathbb{E}}_{V\sim\xi}\big\|u_i^\top V u_i\big\|_2^2$$

$$= \max_{\substack{u_1,\ldots,u_d\in\mathbb{R}^n \\ \text{orthonormal}}}\mathop{\mathbb{E}}_{V\sim\xi}\big\|\sum_{i=1}^{d}u_i^\top V u_i\big\|_2^2$$

$$= \max_{\substack{\Phi=[u_1,\ldots,u_d] \\ \Phi^\top\Phi=I}}\mathop{\mathbb{E}}_{V\sim\xi}\big\|\Phi\Phi^T V\big\|^2$$

$$= \max_{\phi\in\mathcal{R}}\mathop{\mathbb{E}}_{V\sim\xi}\big\|\Pi_\phi V\big\|_2^2.$$

Which gives the desired result. The equivalence with the eigenvectors of $\mathbb{E}_\xi VV^\top$ follows from writing

$$\mathop{\mathbb{E}}_{V\sim\xi}(u^\top V)_2^2 = \mathop{\mathbb{E}}_{V\sim\xi}u^\top VV^\top u$$
$$= u^\top\mathop{\mathbb{E}}_{V\sim\xi}\big[VV^\top\big]u$$

and appealing to a Rayleigh quotient argument, since we require $u_i^*$ to be of unit norm.

# F  The Optimization Problem (2) as a Quadratic Program

**Proposition 1.** *The optimization problem (2) is equivalent to a quadratic program with quadratic constraints.*

*Proof.* For completeness, let $n$, $d$ be the number of states and features, respectively. We consider representations $\Phi \in \mathbb{R}^{n \times d}$. Recall that $\Pi_\phi$ is the projection operator onto the subspace spanned by $\Phi$, that is

$$\Pi_\phi = \Phi(\Phi^\top \Phi)^{-1} \Phi^\top.$$

We will also write $\mathcal{P}_d$ for the space of deterministic policies. We write (2) in epigraph form (Boyd and Vandenberghe, 2004):

$$\text{min. } \max_\pi \left\|\Pi_\phi V^\pi - V^\pi\right\|_2^2 \Leftrightarrow$$

$$\text{min. } \max_{\pi \in \mathcal{P}_d} \left\|\Pi_\phi V^\pi - V^\pi\right\|_2^2 \Leftrightarrow$$

$$\text{min. } t \quad \text{s.t.} \left\|\Pi_\phi V^\pi - V^\pi\right\|_2^2 \leq t \; \forall \pi \in \mathcal{P}_d.$$

The first equivalence comes from the fact that the extremal vertices of our polytope are achieved by deterministic policies. The norm in the constraint can be written as

$$\begin{aligned}
\left\|\Pi_\phi V^\pi - V^\pi\right\|_2^2 &= \left\|(\Pi_\phi - I)V^\pi\right\|_2^2 \\
&= V^{\pi\top}(\Pi_\phi - I)^\top (\Pi_\phi - I)V^\pi \\
&= V^{\pi\top}(\Pi_\phi - I)^\top (\Pi_\phi - I)V^\pi \\
&\overset{(a)}{=} V^{\pi\top}(\Pi_\phi^2 - 2\Pi_\phi + I)V^\pi \\
&\overset{(b)}{=} V^{\pi\top}(I - \Pi_\phi)V^\pi,
\end{aligned}$$

where $(a)$ and $(b)$ follow from the idempotency of $\Pi_\phi$. This is

$$\left\|\Pi_\phi V^\pi - V^\pi\right\|_2^2 = V^{\pi\top}\left(I - \Phi(\Phi^\top \Phi)^{-1}\Phi^\top\right)V^\pi.$$

To make the constraint quadratic, we further require that the representation be left-orthogonal: $\Phi^\top \Phi = I$. Hence the optimization problem (2) is equivalent to

$$\text{minimize } t \quad \text{s.t.}$$
$$V^{\pi\top}(I - \Phi\Phi^\top)V^\pi \leq t \quad \forall \pi \in \mathcal{P}_d$$
$$\Phi^\top \Phi = I.$$

From inspection, these constraints are quadratic. $\qquad\square$

However, there are an exponential number of deterministic policies and hence, an exponential number of constraints in our optimization problem.

# G  NP-hardness of Finding AVFs

**Proposition 2.** *Finding $\max_{\pi \in \mathcal{P}_d} \delta^\top V^\pi$ is NP-hard, where the input is a deterministic MDP with binary-valued reward function, discount rate $\gamma = 1/2$ and $\delta : \mathcal{X} \to \{-1/4, 0, 1\}$.*

We use a reduction from the optimization version of minimum set cover, which is known to be NP-hard (Bernhard and Vygen, 2008, Corollary 15.24). Let $n$ and $m$ be natural numbers. An instance of set cover is a collection of sets $\mathcal{C} = \{C_1, \ldots, C_m\}$ where $C_i \subseteq [n] = \{1, 2, \ldots, n\}$ for all $i \in [m]$. The minimum set cover problem is

$$\min_{\mathcal{J} \subseteq [m]} \left\{ |\mathcal{J}| : \bigcup_{j \in \mathcal{J}} C_j = [n] \right\}.$$

Given a Markov decision process $\langle \mathcal{X}, \mathcal{A}, r, P, \gamma \rangle$ and function $\delta : \mathcal{X} \to [-1, 1]$ define

$$R(\pi) = \sum_{x \in \mathcal{X}} \delta(x) V^\pi(x) \,.$$

We are interested in the optimization problem

$$\max_{\pi \in \mathcal{P}_d} R(\pi) \,. \tag{10}$$

When $\delta(x) \geq 0$ for all $x$ this corresponds to finding the usual optimal policy, which can be found efficiently using dynamic programming. The propositions claims that more generally the problem is NP-hard.

Consider an instance of set cover $\mathcal{C} = \{C_1, \ldots, C_m\}$ over universe $[n]$ with $m > 1$. Define a deterministic MDP $\langle \mathcal{X}, \mathcal{A}, r, P, \gamma \rangle$ with $\gamma = 1/2$ and $n + m + 2$ states and at most $m$ actions. The state space is $\mathcal{X} = \mathcal{X}_1 \cup \mathcal{X}_2 \cup \mathcal{X}_3$ where

$$\mathcal{X}_1 = \{u_1, \ldots, u_n\} \qquad \mathcal{X}_2 = \{v_1, \ldots, v_m\} \qquad \mathcal{X}_3 = \{g, b\} \,.$$

The reward function is $r(x) = \mathbb{I}_{[x=g]}$. The transition function in a deterministic MDP is characterized by a function mapping states to the set of possible next states:

$$N(x) = \bigcup_{a \in \mathcal{A}} \{x' : P(x' \mid x, a) = 1\} \,.$$

We use $\mathcal{C}$ to choose $P$ as a deterministic transition function for which

$$N(x) = \begin{cases} \{x\} & \text{if } x \in \mathcal{X}_3 \\ \{g, b\} & \text{if } x \in \mathcal{X}_2 \\ \{v_j : i \in C_j\} & \text{if } x = u_i \in \mathcal{X}_1 \,. \end{cases}$$

This means the states in $\mathcal{X}_3$ are self transitioning and states in $\mathcal{X}_2$ have transitions leading to either state in $\mathcal{X}_3$. States in $\mathcal{X}_1$ transition to states in $\mathcal{X}_2$ in a way that depends on the set cover instance. The situation is illustrated in Figure 5. Since both policies and the MDP are deterministic, we can represent a policy as a function $\pi : \mathcal{X} \to \mathcal{X}$ for which $\pi(x) \in N(x)$ for all $x \in \mathcal{X}$. To see the connection to set cover, notice that

$$\bigcup_{v_j \in \pi(\mathcal{X}_1)} C_j = [n] \,, \tag{11}$$

where $\pi(\mathcal{X}_1) = \{\pi(x) : x \in \mathcal{X}_1\}$. Define

$$\delta(x) = \begin{cases} 1 & \text{if } x \in \mathcal{X}_1 \\ -1/4 & \text{if } x \in \mathcal{X}_2 \\ 0 & \text{if } x \in \mathcal{X}_3 \,. \end{cases}$$

Using the definition of the value function and MDP,

$$\begin{aligned} R(\pi) &= \sum_{x \in \mathcal{X}} \delta(x) V^\pi(x) \\ &= \sum_{x \in \mathcal{X}_1} V^\pi(x) - \frac{1}{4} \sum_{x \in \mathcal{X}_2} V^\pi(x) \\ &= \sum_{x \in \mathcal{X}_1} V^\pi(x) - \frac{1}{4} \sum_{x \in \mathcal{X}_2} \mathbb{I}_{[\pi(x)=g]} \\ &= \frac{1}{2} \sum_{x \in \mathcal{X}_1} \mathbb{I}_{[\pi(\pi(x))=g]} - \frac{1}{4} \sum_{x \in \mathcal{X}_2} \mathbb{I}_{[\pi(x)=g]} \,. \end{aligned}$$

The decomposition shows that any policy maximizing (10) must satisfy $\pi(\pi(\mathcal{X}_1)) = \{g\}$ and $\pi(\mathcal{X}_2 \setminus \pi(\mathcal{X}_1)) = \{b\}$ and for such policies

$$R(\pi) = \frac{1}{2} \left( n - \frac{1}{2} |\pi(\mathcal{X}_1)| \right) \,.$$

In other words, a policy maximizing (10) minimizes $|\pi(\mathcal{X}_1)|$, which by (11) corresponds to finding a minimum set cover. Rearranging shows that

$$\min_{\mathcal{J} \subseteq [m]} \left\{ |\mathcal{J}| : \bigcup_{j \in \mathcal{J}} C_j = [n] \right\} = 2n - 4 \max_{\pi \in \mathcal{P}_d} R(\pi).$$

The result follows by noting this reduction is clearly polynomial time.

Figure 5: The challenging MDP given set cover problem $\{\{1, 2, 3\}, \{1, 4\}, \{4\}\}$. State $g$ gives a reward of $1$ and all other states give reward $0$. The optimal policy is to find the smallest subset of the middle layer such that for every state in the bottom layer there exists a transition to the subset.

## H   Empirical Studies: Methodology

### H.1   Four-room Domain

The four-room domain consists of 104 discrete states arranged into four "rooms". There are four actions available to the agent, transitioning deterministically from one square to the next; when attempting to move into a wall, the agent remains in place. In our experiments, the top right state is a goal state, yielding a reward of 1 and terminating the episode; all other transitions have 0 reward.

### H.2   Learning $\phi$

Our representation $\phi$ consists of a single hidden layer of 512 rectified linear units (ReLUs) followed by a layer of $d$ ReLUs which form our learned features. The use of ReLUs has an interesting side effect that all features are nonnegative, but other experiments with linear transforms yielded qualitatively similar results. The input is a one-hot encoding of the state (a 104-dimensional vector). All layers (and generally speaking, experiments) also included a bias unit.

The representation was learned using standard deep reinforcement learning tools taken from the Dopamine framework (Castro et al., 2018). Our loss function is the mean squared loss w.r.t. the targets, i.e. the AVFs or the usual value function. The losses were then trained using RMSProp with a step size of 0.00025 (the default optimizer from Dopamine), for 200,000 training updates each over a minibatch of size 32; empirically, we found our results robust to small changes in step sizes.

In our experiments we optimize both parts of the two-part approximation defined by $\phi$ and $\theta$ simultaneously, with each prediction made as a linear combination of features $\phi(x)^\top \theta_i$ and replacing $\tilde{L}(\phi; \boldsymbol{\mu})$ from (5) with a sample-based estimate. This leads to a slightly different optimization procedure but with similar representational characteristics.

### H.3   Implementation Details: Proto-Value Functions

Our PVF representation consists in the top $k$ left-singular vectors of the successor representation $(I - \gamma P^\pi)^{-1}$ for $\pi$ the uniformly random policy, as suggested by Machado et al. (2018); Behzadian and Petrik (2018). See Figure 9 for an illustration.

## H.4 Learning AVFs

The AVFs were learned from 1000 policy gradient steps, which were in general sufficient for convergence to an almost-deterministic policy. This policy gradient scheme was defined by directly writing the matrix $(I - \gamma P^\pi)^{-1}$ as a Tensorflow op (Abadi et al., 2016) and minimizing $-\delta^\top (I - \gamma P^\pi)^{-1} r$ w.r.t. $\pi$. We did not use an entropy penalty. In this case, there is no approximation: the AVF policies are directly represented as matrices of parameters of softmax policies.

## H.5 SARSA

In early experiments we found LSPI and fitted value iteration to be somewhat unstable and eventually converged on a relatively robust, model-based variant of SARSA.

In all cases, we define the following dynamics. We maintain an occupancy vector $d$ over the state space. At each time step we update this occupancy vector by applying one transition in the environment according to the current policy $\pi$, but also mix in a probability of resetting to a state uniformly at random in the environment:

$$d = 0.99 d P^\pi + 0.01 \text{Unif}(\mathcal{X})$$

The policy itself is an $\epsilon$-greedy policy according to the current $Q$-function, with $\epsilon = 0.1$.

We update the $Q$-function using a semi-gradient update rule based on expected SARSA (Sutton and Barto, 1998), but where we simultaneously compute updates across all states and weight them according to the occupancy $d$. We use a common step size of 0.01 but premultiplied the updates by the pseudoinverse of $\Phi^\top \Phi$ to deal with variable feature shapes across methods. This process was applied for 50,000 training steps, after which we report performance as the average value and/or number of steps to goal for the 10 last recorded policies (at intervals of 100 steps each).

Overall, we found this learning scheme to reduce experimental variance and to be robust to off-policy divergence, which we otherwise observed in a number of experiments involving value-only representations.

# I   Representations as Principal Components of Sets of Value Functions

In the main text we focused on the use of value functions as auxiliary tasks, which are combined into the representation loss (5). However, Section 3.2 shows that doing so is equivalent (in intent) to computing the principal components of a particular set of value functions, where each "column" corresponds to a particular auxiliary task.

In Figure 10 we show the representations generated from this process, using different sets of value functions. For completeness, we consider:

- 1000 AVFs,
- 1000 random deterministic policies (RDPs),
- 1000 random stochastic policies (RSPs), and
- The 104 rows of the successor matrix (corresponding to proto-value functions).

As with principal component analysis, the per-state feature activations are determined up to a signed scalar; we pick the vector which has more positive components than negative. In all but the PVF case, we sample a subset of the many possible value functions within a particular set. Figure 11 shows that the AVF approach is relatively robust to the sample size.

The AVFs are sampled using Algorithm 1, i.e. by sampling a random interest function $\delta \in [-1, 1]^n$ and using policy gradient on a softmax policy to find the corresponding value function. The random policies were generated by randomly initializing the same softmax policy and using them as-is (RSPs) or multiplying the logits by 1e6 (RDPs).

Figure 6: Figure 2, enlarged. Red arrows highlight states where $\delta$ and $d_\pi$ have opposite signs.

Figure 7: Four interest functions sampled from $\{-1, 1\}^n$, along with their corresponding flow $d_\pi$, adversarial value function, and corresponding policy. The top example was chosen to illustrate a scenario where $d_\pi(x) < 0$ but $V^\pi(x) > 0$; the other three were selected at random. In our experiments, sampling from $[-1, 1]^n$ yielded qualitatively similar results.

Figure 8: 16-dimensional representations learned by training a deep network to predict the value function of a single policy, namely: the uniformly random policy, the optimal policy, and a convex combination of the two in equal proportions.

Figure 9: 16-dimensional representation generated by the proto-value function method (Mahadevan and Maggioni, 2007) applied to left-singular vectors of the transition function corresponding to the uniformly random policy. The top-left feature, labelled '1', corresponds to the second largest singular value. Notice the asymmetries arising from the absorbing goal state and the walls.

Figure 10: 16-dimensional representations generated from the principal components of different sets of value functions. Beginning in the top-left corner, in clockwise order: from $k = 1000$ AVFs sampled according as in 1; proto-value functions (9); from $k = 1000$ random deterministic policies (RDPs); and finally from $k = 1000$ random stochastic policies. Of the four, only PVFs and AVFs capture the long-range structure of the four-room domain.

Figure 11: 16-dimensional representations generated from the principal components of sets of AVFs of varying sizes ($k = 20, 100, 400, 1000$). To minimize visualization variance, each set of AVFs contains the previous one. The accompanying video at `https://www.youtube.com/watch?v=q_XG7GhImQQ` shows the full progress from $k = 16$ to $k = 1024$.

## Random policies

## Corresponding value functions

Figure 12: A sample of random deterministic policies, together with their corresponding value functions. These policies are generated by assigning a random action to each state. Under this sampling scheme, it is unlikely for a long chain of actions to reach the goal, leading to the corresponding value functions being zero almost everywhere.

Figure 13: Average return (left) and average steps to goal (right), achieved by policies learned using a representation, with given number of features, produced by VALUE, AVF, or PVF. Average is over all states and 20 random seeds, and shading gives standard deviation.