[Reviews · NeurIPS 2019]

Reviewer 1



This paper is very intriguing. Although there is no conclusive empirical evidence of the usefulness of auxiliary tasks, their design and justification remain on the whole ad-hoc. This paper describes a new method based on geometric properties of the space of value functions to represent learning. The results show that predicting adversarial value functions as auxiliary tasks leads to rich representations. Overall, this innovative perspective to represent learning is good for us to understand the learning process. and the literature review shows that the author is knowledgeable in this field. As the author said as I quote, their work may “opens up the possibility of automatically generating auxiliary tasks in deep reinforcement learning ”. Here are my major concerns: The author tries to describe his representation starting at part 2. A description of the previous version of representation would be better for a reviewer to get a general idea before describing a new representation for RL. I would suggest the author put “Related work” in part 2 instead of part 5.

Reviewer 2



This paper studies the problem of learning useful representations for reinforcement learning through the lens of an adversarial framework. In particular, a good representation is identified as one that yields low linear value-function estimation error if an adversary is able to choose a value function (induced by a policy). The paper shows first that the the only policies that should be considered are deterministic, and then identifies a more narrowed set of adversarial values, though the number is still exponential. I really liked the theoretical insights of this paper, and because of this I tend to vote for acceptance, though I claim that experiments are too preliminary. Some more comments below: 1- in (1) highlight more clearly that \phi is the only optimization knob. 2- in terms of readability, it is unclear why Lemma 1 is useful until after i read the proof of Theorem 1 from the Appendix. Maybe consider saying why this Lemma is useful, or move things around 3- isn't the first half of Lemma 1 (solution lies in the set of extreme points) a very well-known result in linear programming? If yes, then be more clear that this is not new. 4- this adversarial framework reminds me a lot of the use of Wasserstein distance in model-based RL, whereby a good model is defined as one that yields low error in the context of adversarial choice of value functions (that are Lipschitz). Do you see a synergy here? is there any deep connection? Also, can you clarify why you used model-based algorithms in experiments? There is no mention of model-based stuff until we get to experiments, so i am wondering if there is a connection. 5- for proof of Theorem 2 in appendix, maybe do define idempotent matrices and their properties. I checked the proofs of the first two theorems and otherwise they seem sound and clear. 6- the part that the paper falls short is experiments. It could still be OK if the authors showed a clear path towards extending the idea to function approximation, but this is lacking. Plus, the method cannot really beat the baseline even in the toy domain. Any comment on challenges when going to function approximation? ---- post rebuttal: I am happy to see that the authors are willing to add a section that more seriously tackles/starts to think about challenges when going to arbitrary function approximators in practice. As for the point about a potential model-based RL result, Farahmand and friends was indeed the paper that I had in mind. Also, because of the focus on linearity, Parr and friends 2008 on linear models shows a deeper connection/equivalence, and so could be useful. It would be very neat if there was a deeper connection. If one cannot be shown in this paper, can a conjecture still be made?

Reviewer 3



The paper claims that the best representation should minimize the maximum error in approximating all the possible value functions and not a single one that pertains to a given policy. As such, the authors establish further results, which reduce the maximization over all policies to the maximization over the finite set of external vertices which corresponds to (distinct in Lebesgue measure) deterministic policies. These value functions are termed adversarial (although these are adversarial only from approximation point-of-view). I enjoyed reading the paper. Here are some comments, which may help improving the paper. The presented theory is indeed linear, but it is coated as being generic through some introduction on the representation mapping \phi and the so-called "two-part approximation". The distinction is that in a generic case, \phi becomes part of the computation through its gradient and such, which is not the case here. I am personally not favouring the presented perspective, yet it is not a real issue to argument against. Note that in all classic sources (e.g. Bertsekas and Tsitsiklis numerous books and papers) V = \Phi \theta is the definition of linear ADP. There is no need for saying otherwise. Figure 1 (Right): I would add the axis. The current figure is hard to understand. Is it the value space for a 2 dimensional state-space? i.e., V^{pi}(x(1)) vs V^{pi}(x(2)) ? L94 --> \mathcal{V} definition: looks like it should be “for all” not “for some”. Additionally, mathematical statements should be succinct. I would remove it if you mean “for all”. Supplementary material -> the first equation after equation 6: the sub-scripts of v’s should be super-script. L107: \phi should be in R^d and not \mathcal{R}=R^{nxd}. In other words, \phi is a row of \Phi. \delta is called interest function in section 3.1, yet it was called direction in Lemma 1.

[Author Response · NeurIPS 2019]

We thank the reviewers for their highly useful feedback, which we will incorporate in the revised version. To summarize, the main points are:

- Highlight relationship with existing representation formalisms
- Clarify how the ideas can be extended to practical settings

# 1 Reviewer 1

**Put "Related work" in part 2.** We agree we can improve the connection with other formalisms, we will emphasize this.

# 2 Reviewer 2

**Isn't the first half of Lemma 1 a well known result?** You are right in that the proof makes direct use of existing linear programming results. There is a slight difference, however, since Lemma 1 is specifically about the value polytope: the domain of the functional is nonconvex, and the proof requires first an extension to its convex hull. The statement regarding deterministic policies is also polytope-specific. We will make sure to emphasize what is novel in the statement.

**Consider saying why Lemma 1 is useful**. The usefulness is the interesting property of the value polytope (which we then make use of). We will emphasize this point also.

**Connection to Wasserstein distance and model-based RL.** In that space, we know of Farahmand, Barreto, Nikovski (2017) and follow-up work. This is an interesting connection, although to the best of our knowledge there is no model learning equivalent of the value polytope or adversarial value functions. E.g. we would need a small set of models w.r.t. which we want low modelling error. Please let us know in your revised review if there are specific papers that might have a closer connection.

**Can you clarify why you used model-based algorithms in experiments?** Model-free experiments require dealing with stochasticity in the results, error bars, etc. and would give a murkier picture of the role of the representation. We could learn AVFs using Monte-Carlo samples from the different adversarial policies, and learn these policies through sample-based policy gradient, but we feel our setup gets to the point more clearly. We will highlight this.

**The experiments are preliminary / challenges when going to function approximation.** To clarify, our experiments use tabular information (e.g., $V^\pi$) to produce function approximation, so we read this comment as "how to learn a representation when tabular information is not available".

The path to a "deep" method is clear to us, but not short. There are a few challenges to overcome: 1) how should we represent adversarial policies? 2) what is the effect of non-uniform distributions on the learned representation? this effectively changes the norm in Equation 1; 3) what is the effect of changing the sampling distribution for $\delta$? and 4) we would like to make use of Bellman updates to learn AVFs, but these requires a good off-policy learning method. 1–3 might be answered by considering the auxiliary tasks perspective (Section 3.2). We will add a section discussing these points.

# 3 Reviewer 3

$\phi$ **becomes part of the computation through its gradient.** This is a fair point. At first glance the difference in our perspectives seems to be about how the optimization process discovers the representation, rather than the space of possible representations, but if you are making a more specific distinction please let us know in the revised review. Either way, we will make sure to discuss this point.

**Figure 1 (Right)** is a cartoon version of a value polytope for a 2-state MDP. The axes are a good suggestion.

Thank you for catching bugs in the math, we will fix.

[Meta-Review · NeurIPS 2019]

The paper studies representation learning for RL through a notion of adversarial value functions. This provides a new conceptual/theoretical understanding of representation learning for RL that all reviewers felt was interesting. The experimental results are somewhat preliminary, but sufficient.